# Pathophysiology of Atopic Dermatitis and Psoriasis: Implications for Management in Children

**DOI:** 10.3390/children6100108

**Published:** 2019-10-04

**Authors:** Raj Chovatiya, Jonathan I. Silverberg

**Affiliations:** 1Department of Dermatology, Northwestern University Feinberg School of Medicine, Chicago, IL 60611, USA; raj.chovatiya@gmail.com; 2Department of Dermatology, George Washington University School of Medicine, Washington, DC 20037, USA

**Keywords:** atopic dermatitis, psoriasis, inflammation, cytokine, skin-barrier, topical therapy, systemic therapy, biologic therapy

## Abstract

Atopic dermatitis (AD) and psoriasis are chronic inflammatory skin diseases associated with a significant cutaneous and systemic burden of disease as well as a poor health-related quality of life. Here, we review the complex pathophysiology of both AD and psoriasis and discuss the implications for treatment with current state-of-the-art and emerging topical and systemic therapies. Both AD and psoriasis are caused by a complex combination of immune dysregulation, skin-barrier disruption, genetic factors, and environmental influences. Previous treatments for both diseases were limited to anti-inflammatory agents that broadly suppress inflammation. Emerging insights into relevant pathways, including recognition of the role of T-helper type 2 driven inflammation in AD and T-helper 1 and 17 driven inflammation in psoriasis, have led to a therapeutic revolution. There are a number of novel treatment options available for AD and psoriasis with many more currently under investigation.

## 1. Introduction

Atopic dermatitis (AD) and psoriasis are chronic, inflammatory skin diseases associated with considerable morbidity. Though both AD and psoriasis were once considered to be skin-limited disorders, emerging evidence suggests a substantial systemic burden of disease. AD is associated with higher rates of mental health symptoms and disorders, including depression, anxiety, and attention-deficit (hyperactivity) disorder, sleep dysregulation, other atopic disorders (e.g., asthma, hay fever), cardiovascular disease, stroke, and obesity [1,2,3,4,5,6,7]. Psoriasis is associated with rheumatologic (psoriatic arthritis), cardiovascular, metabolic, hepatic, and psychiatric disease [8,9,10,11]. Both AD and psoriasis are strongly associated with poor health-related quality of life (QOL), high direct and indirect costs of care, and significant societal cost—highlighting a need for optimal disease control [12,13,14,15].

AD and psoriasis are caused by a complex interplay between skin-barrier disruption, immune dysregulation, host genetics, and environment triggers [16,17]. Both diseases result in chronic, systemic inflammation with increased circulating populations of leukocytes, lymphocytes, cytokines, and chemokines (predominantly Th2 pathways in AD vs. Th1 and Th17 pathways in psoriasis) [18,19]. Previously, therapeutic options for AD and psoriasis, particularly extensive and/or severe disease, were limited to topical and non-targeted systemic immunosuppressants that had poor efficacy, safety and/or tolerability. In recent years, a better understanding of the underlying pathologic mechanisms in both conditions has led to a revolution in the development of novel, targeted therapies.

This manuscript reviews the pathophysiology of AD and psoriasis, current and novel therapeutics, and implications for management in children.

## 2. Atopic Dermatitis

Atopic dermatitis affects up to 15–20% of children and 1–10% of adults worldwide [20]. Approximately two-thirds of affected children present by two years of age, and 80% present by five years of age [21]. AD is a heterogeneous disorder associated with variable symptoms (e.g., pain, sleep disturbance, depression, anxiety), signs (e.g., papules, plaques, xerosis, erosions, oozing/weeping, lichenification, prurigo nodules, lichenoid papules, follicular accentuation), distributions (e.g., flexural, extensor, head and neck, hands, trunk), age of onset (e.g., pediatric, adult) and persistence (e.g., transient, intermittent, persistent) [4,5,22,23,24,25,26,27,28,29,30,31,32,33,34]. While some clinical characteristics are more common in children (e.g., ventral wrist dermatitis), others are more common in adults (e.g., hand and foot dermatitis, nipple eczema, thinning of the eyebrows, modification of disease course by emotional or environmental factors) [24].

AD is characterized by dysfunction in both non-immune and immune components of the skin-barrier. Despite this general framework, there is continued debate as to the exact etiology. The “outside-in” hypothesis posits that physical changes in the epidermal structure (e.g., keratinocytes, proteins, lipids, pH) drive immune dysregulation, whereas the “inside-out” hypothesis suggests that aberrant immune activity (e.g., leukocytes, lymphocytes, cytokines, antimicrobial peptides) alters the skin-barrier [35]. Despite the incongruity in the sequence of events, both models agree upon many of the same common mechanisms that result in chronic inflammation.

### 2.1. Pathophysiology

#### 2.1.1. The Outside-In Hypothesis

The physical barrier function of the epidermis is crucial for protection against pathogens, allergens, toxins, and other irritants and maintenance of appropriate skin hydration. According to the outside-in hypothesis, a compromise in the complex matrix of epidermal keratinocytes, proteins, and/or lipids predisposes the epidermis to external insults, transepidermal water loss (TEWL) and reduced skin hydration, resulting in immune dysregulation and inflammation [36]. Previous studies found that lesional skin from AD patients had significant barrier dysfunction compared to healthy controls. Moreover, even previously inflamed and non-lesional skin continues to show barrier disruption [37].

Further evidence for importance of barrier dysfunction in AD is provided by loss-of-function mutations in the filaggrin gene (FLG) [38,39]. The outermost skin-barrier is composed of the cornified envelope, a highly cross-linked, flexible, insoluble structure composed of corneocytes, lipids, involucrin, loricrin, and filaggrin. Filaggrin, cleaved from its precursor profilaggrin, plays a key role in aligning and securing corneocytes, protein fibrils, and lamellar sheets [40]. Loss-of-function mutations in FLG confer increased risk of developing AD, earlier onset, persistent disease, other atopic disorders, and infection [41,42,43]. The processing of profilaggrin to filaggrin produces polycarboxylic acids (e.g., urocanic acid, pyrrolidone-5-carboxylic acid) known as natural moisturizing factors (NMFs) These NMFs promote hydration of the skin. A decrease in NMFs as a result of impaired profilaggrin cleavage decreases osmotic draw and results in increased TEWL [44].

NMFs and other acidic metabolites, including free fatty acids, function more broadly in decreasing skin surface pH [45]. Acidic skin pH (4–6) is important for appropriate expression, confirmation, and modification of epidermal proteins [46], inhibition (e.g., proteases) and activation (e.g., lipases) of enzymes that are critical for barrier function [47,48], and regulation of nonpathogenic skin flora [49]. Patients with AD have significantly higher pH in both lesional and non-lesional skin [50,51].

Tight junctions are epidermal protein complexes composed of claudins and occludins that are designed to seal off intercellular space, control permeability to solutes, water, pathogens, antigens, and allergens, and maintain cell polarity [52]. Functional impairment of tight junctions has been observed in patients with AD and results in skin-barrier dysfunction [53,54,55].

A variety of external and environmental factors have also been shown to contribute to skin-barrier impairment in AD patients, including in utero exposures [56], bacterial dysbiosis [57], harsh climate [58], water hardness [59,60], airborne pollutants [61,62,63], tobacco smoke [64], personal care products (i.e., irritants and pruritogens) [65,66,67], and, in certain cases, contact allergens [68]. Kantor and Silverberg have recently reviewed the topic of environmental risk factors in AD [69].

Thus, a variety of factors contribute to dysfunction of the physical epidermal barrier, resulting in keratinocyte damage, enhanced antigen and allergen penetration, and increased microbial signaling, ultimately resulting in immune cell activation, cytokine production, and inflammation.

#### 2.1.2. The Inside-Out Hypothesis

According to the inside-out hypothesis of AD, dysregulation of the immune system results in disruption of the keratinocyte barrier and inflammation [70]. General support for this model is provided by the fact that many patients with AD have genetic polymorphisms in cytokines, chemokines, and receptors (including IL-4, IL-13, IL-18, IL-22, IL-31, CCL5, and CD14) [71,72,73,74], and treatment of human epidermal keratinocytes with many of these same mediators recapitulates the atopic dermatitis phenotype [75].

The innate immune system is composed of soluble and cellular effectors that utilize germline-encoded pattern recognition receptors (PRRs). Genetic polymorphisms in Toll-like receptors (TLRs), one such family of PRRs, confers increased risk of AD and staphylococcus aureus (SA) infection [76,77]. Antimicrobial peptides (AMPs), a diverse group of soluble, amphipathic, cationic proteins, are decreased in the skin of atopics and associated with increased TEWL, fatty acid loss, increased pH, and colonization with SA [78,79,80]. Various subsets of CD11c+ and CD1a+ dendritic cells (DCs), which specialize in antigen uptake and presentation to lymphocytes, are increased in number and activity in patients with AD and contribute to T-cell activation and polarization [81,82]. Type 2 innate lymphoid cells (ILCs) also show increased activity in AD patients and contribute to DC activation by producing IL-5 and IL-13 [83]. Keratinocyte-derived IL-25, IL-33, and thymic stromal lymphopoietin (TSLP) are increased in atopic skin and contribute to activation of DCs, type 2 ILCs, and type 2 inflammation. Though eosinophils and mast cells can both be elevated in the skin and blood of AD patients, they likely play a lesser role in the pathogenesis of AD [84].

The adaptive immune system is composed of antigen-specific lymphocytes (T-cells, B-cells) that utilize long-lasting, specific memory to direct the host immune response. Previously, AD was thought to be a disease of Th1/Th2 imbalance; however, more recent data suggest a biphasic T-cell response with additional roles played by Th17 and Th22 cells [85]. All stages of AD (early through late) are dominated by Th2 cell activity, as evidenced by increased production of the type 2 cytokines and chemokines IL-4, IL-5, IL-10, IL-13, IL-31, and CCL5 [86]. This group of mediators is involved in allergic responses, B-cell class-switching to IgE, impairment of terminal keratinocyte differentiation (through inhibition of filaggrin, loricrin, and involucrin), and downregulation of AMPs, all of which lead to increased skin permeability to exogenous antigens and pathogens [75,87]. Later stages of AD show upregulation of Th1 activity, resulting in higher levels of IFN-γ and ultimately keratinocyte apoptosis [88]. Th17 cells, a subset of T-cells defined by production of IL-17 and IL-22, appear to be increased in number but decreased in activity in AD (unlike psoriasis). Their role in pathogenesis still remains unclear. Th22 cells, a more recently described subset of T-cells that show an increased number and activity in AD lesions, produce high levels of IL-22 that may induce epidermal hyperplasia and terminal keratinocyte differentiation [89].

### 2.2. Treatments

#### 2.2.1. Topical Treatments

Topical therapies are typically the first-line treatment modality for AD. They are often used as monotherapy in mild to moderate disease and in combination with systemic, biologic, or phototherapy in moderate to severe cases. Moisturizers and emollients restore the dysfunctional epidermal barrier by reducing TEWL and in certain formulations, replacing and supplementing lipids [90]. Emollients include thicker, occlusive vehicles like petrolatum and thinner, more spreadable vehicles such as creams and lotions, which are generally less effective sealants. Topical corticosteroids (TCS) act broadly to suppress the activity of innate and adaptive immune cells. Numerous randomized controlled trials (RCTs) demonstrated efficacy for both reactive and proactive treatment protocols [91]. While TCS are overall safe, efficacious, and well tolerated for a wide spectrum of disease, adverse effects (AEs) of their chronic use include skin atrophy, fragility, striae, poor wound healing, local immunosuppression, and potential systemic absorption leading to adrenal insufficiency. Topical calcineurin inhibitors (TCIs), including tacrolimus and pimecrolimus, inhibit calcineurin signaling thereby inhibiting T-cell activation. TCIs are most often used in areas of thinner skin (e.g., eyelids, axillae, genitals) that have a higher risk of AEs from TCS. They are less commonly used as first-line therapy owing to high cost, activity comparable to low potency TCS [92], slower onset than TCS, application site reactions (e.g., stinging, burning) and an FDA-issued black box warning for potential malignancy. Long-term manufacturer sponsored registries for pimecrolimus and tacrolimus have found no evidence of increased malignancy to-date [92]. As such, TCIs are still recommended for topical treatment of AD in infants, children and adults [91].

#### 2.2.2. Systemic Treatments

Systemic corticosteroids, unlike TCS, universally suppress immune activity. While effective in the short-term and as a bridge therapy, their long-term use is limited by a poor AE profile including hyperglycemia, diabetes, weight gain, insomnia, cardiovascular disease, osteoporosis and osteonecrosis, adrenal insufficiency, gastroesophageal reflux, and immune suppression [93,94]. Systemic immunosuppressants, including cyclosporine A, methotrexate, azathioprine, and to a lesser extent, mycophenolate mofetil, demonstrated efficacy for moderate to severe AD in a variety of RCTs [95]. While more specific than systemic corticosteroids, given their relatively broad effects on immune function, systemic immunosuppressants still have a broad AE profile and are often poorly tolerated and not ideal options for long-term treatment.

#### 2.2.3. Phosphodiesterase-4 Inhibition

In 2016, the FDA approved the first topical phosphodiesterase-4 (PDE-4) inhibitor (crisaborole) for use in mild-to-moderate AD in children and adults. PDE-4 is responsible for degrading cAMP in immune cells, which promotes the expression of pro-inflammatory cytokines like IFN-γ and IL-17 and inhibits production of anti-inflammatory cytokines such as IL-10 [96]. Two phase III, double-blind, placebo controlled RCTs in adults and children showed that crisaborole was significantly more effective than vehicle with no major AEs other than application site reactions (e.g., stinging) [97]. Other topical PDE-4 inhibitors are currently in phase I, II, and II development [98]. Apremilast, an oral PDE-4 inhibitor that is approved for the treatment of moderate-to-severe psoriasis, has also been studied in AD. A recent phase II showed no difference compared to vehicle at lower doses and modest (but statistically significant) improvement in AD at higher doses [99]. However, higher doses of apremilast were associated with a high rate of treatment emergent AEs resulting in its discontinuation during the trial [99].

#### 2.2.4. IL-4 And IL-13 Inhibition

In 2017, the FDA approved dupilumab, a fully human monoclonal antibody for the treatment of AD. Dupilumab targets IL4Ra, a receptor subunit protein shared by both type I and type II receptors for IL-4 and/or IL-13. In phase II and III double-blind, placebo controlled RCTs, patients with moderate to severe AD treated with dupilumab showed significant improvement in a number of primary and secondary outcomes including clinician reported outcomes (e.g., Eczema Area and Severity Index (EASI), SCORing Atopic Dermatitis (SCORAD), body surface area (BSA) involvement, investigators global assessment (IGA)) and patient-reported outcomes (e.g., pruritus numeric rating scale (NRS), Patient-Oriented Eczema Measure (POEM), and Dermatology Life Quality Index (DLQI)) [100,101,102,103]. Dupilumab demonstrated an excellent safety profile with mainly mild or moderate AEs, including injection site reactions and ocular AEs (dry eye, eye pruritus, blepharitis, conjunctivitis, keratitis), with no increases of infection or malignancy overall, and lower rates of bacterial skin infection compared to placebo.

There are currently ongoing studies examining the impact of monoclonal antibodies targeting IL-13 (lebrikizumab and tralokinumab), which are even more specific than targeting IL4 and IL13 for AD [104]. Lebrikizumab in combination with TCS showed promising results in a phase II study with significant improvement in the proportion of patients achieving ≥50% reduction in EASI from baseline (EASI-50) at 12 weeks compared to placebo [105]. Tralokinumab in combination with mid-potency TCS was effective in a phase II RCT in adults and showed significant improvement in EASI, DLQI, SCORAD, and NRS-itch scores with a significant proportion of patients achieving EASI-50 and EASI-75 at 12 weeks compared to placebo [106].

#### 2.2.5. Il-31 Inhibition

Nemolizumab, a humanized monoclonal antibody targeting the IL-31 receptor (a major signaling pathway for itch in AD), is currently in development. A phase I double-blind, placebo controlled RCT showed that a single dose of nemolizumab in AD patients resulted in significant improvement in visual analog scale (VAS)-itch, sleep and decreased use of TCS compared to placebo [107]. A phase II double-blind placebo controlled RCT demonstrated that nemolizumab 0.5 mg/kg every four weeks significantly improved VAS-itch, EASI, and IGA in adults with AD [108]. The results of phase III studies are pending.

#### 2.2.6. Janus Kinase Inhibition

IL-4, IL-13, IL-31, and other important cytokines in the pathogenesis of AD signal through extracellular receptors that activate intracellular Janus kinases (JAKs) [109,110]. Activation of JAKs results in downstream activation of mitogen-activated protein kinases (MAPKs), phosphoinositide 3-kinases (PI3Ks), and signal transducers and activators of transcription (STATs). Multiple JAK inhibitors are currently approved therapy for treatment of rheumatoid arthritis, psoriatic arthritis, and ulcerative colitis. Emerging data showed potential efficacy for AD. A case series of six adults with moderate to severe AD treated off-label with tofacitinib (an oral JAK 1 and 3 inhibitor) showed significant improvements in pruritus and mean SCORAD [111]. A phase II double-blind, placebo controlled RCT of adults with mild-to-moderate AD demonstrated that topical tofacitinib ointment versus vehicle resulted in significant reduction in EASI, BSA, and Physician’s Global Assessment (PGA). A number of other topical and/or oral JAK inhibitors are currently under investigation for treatment of AD. A phase II double-blind, placebo controlled RCT with baricitinib (an oral selective JAK 1 and 2 inhibitor) in combination with TCS showed improvement in inflammation and pruritus with significantly more patients achieving EASI-50 at 16 weeks compared to placebo [112]. A phase II double-blind, placebo controlled RCT with upadacitinib (an oral JAK 1 inhibitor) monotherapy showed a significant decrease in EASI and NRS itch scores at 16 weeks (primary endpoint) and 32 weeks (long-term extension) [113]. Additional long-term studies are underway to better understand the efficacy and safety profile of JAK inhibitors.

## 3. Psoriasis

Psoriasis is estimated to affect 2–3% of the global population, corresponding to >125 million individuals [114,115]. Though most affected individuals present during adulthood between 18–39 or after the age of 50, approximately one-third of patients experience their first episode of psoriasis under the age of 20 [17,114]. Psoriasis is an immune-mediated disease characterized by cycles of sustained inflammation and remission, uncontrolled keratinocyte proliferation, and impaired keratinocyte differentiation. Inflammation is driven by dysregulation in both the innate and adaptive immunity, and influenced by genetic factors. Plaque psoriasis is characterized by well-demarcated, erythematous, scaly plaques on the trunk and extensor surfaces of the limbs. Inverse psoriasis affects the intertrigenous skin with erythematous, occasionally erosive patches and plaques. Guttate psoriasis is more common in children and adolescents, and is characterized by droplet sized scaly plaques that can be triggered by group A streptococcal infection. Nearly one-half of childhood cases of psoriasis may first present in this way. Pustular psoriasis, occurring in both local and generalized forms, presents with innumerable, coalescing, sterile pustules. While psoriasis can manifest with a variety of morphologies, all subtypes share several important features: (1) hyperplasia of the epidermis (i.e., acanthosis), (2) incomplete keratinization with retention of nuclei (i.e., parakeratosis), (3) newly generated, tortuous superficial blood vessels (i.e., neovascularization), and (4) a dense inflammatory infiltrate composed of DCs, macrophages, neutrophils, and T-cells [116].

### 3.1. Pathophysiology

#### 3.1.1. Dysregulated Immune Activity

The initiation phase of psoriasis consists of an external insult to the epidermis, such as trauma (e.g., Koebner phenomenon), medication (e.g., β-blocker, ACE-inhibitor, lithium), and/or infection (e.g., group A streptococcus) [115]. Following keratinocyte damage, two important events occur to activate the innate immune system: (1) keratinocytes dramatically upregulate the production of AMPs including S100, β-defensins, and LL-37 (i.e., cathelicidin), and (2) damaged keratinocytes release DNA, RNA, and other endogenous danger signals [117,118]. One proposed mechanism for the initiation of psoriasis posits that these AMPs bind to the keratinocyte DNA and RNA and activate plasmacytoid DCs (pDCs), a circulating antigen presenting cell (APC) specialized for recognition of viral pathogens. In particular, LL37 complexes with self-genetic material and binds to TLR7, TLR8, and TLR9, resulting in the secretion of type I interferons (IFN-α and IFN-β) and a break in immunologic tolerance [119,120,121]. Keratinocytes, which also function as a physical barrier and sensory cell in the innate immunity, produce pro-inflammatory cytokines in response to AMP-nucleic acid complexes, including type I IFNs, TNF-α, IL-1, and IL-6 [122]. Type I IFNs promote myeloid DC (mDC) differentiation, activation, and entry into skin-draining lymph nodes [123]. These cells also secrete a number of pro-inflammatory cytokines, such as TNF-α, IL-12, and IL-23, resulting in the differentiation and activation of activation of Th1, Th17, and Th22 cells.

Without specific therapeutic intervention, activation of adaptive immunity drives the self-sustaining, maintenance phase of inflammation in psoriasis [124]. The initial stage of the adaptive immunity is characterized by Th1 cell activity and concomitant production of type 1 cytokines such as IFN-γ, while Th17 cells (and to a lesser extent, Th22 cells) dominate the later portion of the adaptive immune response with production of IL-17, IL-23, IL-21, and IL-22 [125]. Recent studies focused extensively on the role Th17 cells in the pathogenesis of psoriasis, in part due to their high concentration in psoriatic lesions and significant reduction following anti-TNF-α treatment [126]. Th17 cells are a subset of T-lymphocytes that specialize in epithelial surveillance and defense against extracellular pathogens through production of IL-17, IL-23, and IL-22. The IL-17 family is composed of six members, two of which (IL-17A and IL-17F) appear to be most relevant in psoriasis. IL-17 activates a variety of intracellular kinase cascades in effector cells, including JAK, MAPK, NF-κB, and glycogen synthase kinase 3 β (GSK-3-β) [127]. IL-22 and IL-23 link immune activity and keratinocyte function by inducing keratinocyte proliferation and secretion of AMPs and cytokines [128].

Though TNF-α, IL-17, and IL-23 play a key role in the general pathophysiology of psoriasis, especially plaque psoriasis, there are additional relevant pathways in other variants of psoriasis. IL-1, IL-8, and IL-36 appear to be important for neutrophilic activity in pustular psoriasis [129,130], whereas guttate psoriasis may be driven by streptococcal superantigen mediated activation of TCRs and subsequent molecular mimicry of keratin proteins [131,132].

#### 3.1.2. Genetic Factors

Early epidemiologic studies suggested a genetic component to psoriasis, as patients with psoriasis had higher incidence of disease among first and second-degree relatives compared to the general population, and monozygotic twins had two to threefold higher risk of psoriasis compared to dizygotic twins [133,134,135]. These findings were later confirmed by genome-wide linkage studies, which have identified over 70 different genetic loci associated with psoriasis [136,137]. However, variability in these loci are thought to only account for about 30% of heritability, suggesting a cumulative effect from multiple mutations with smaller individual effects and/or undetected gene-gene interactions. The nine most highly associated loci are named psoriasis susceptibility 1 through 9 (PSORS1 through PSORS9), with PSORS1 accounting for 35–50% of heritability in psoriasis [136,138]. PSORS1 is located in the major histocompatibility complex (MHC) on chromosome 6p21, spanning a 220 kb segment class I telomeric region of human leukocyte antigen B (HLA-B). This region contains several candidate genes, including *HLA-C* (associated variant HLA-Cw6 [HLA-C*06:02]; an MHC class I protein), *CDSN* (associated variant allele 5; corneodesmin, a protein expressed in the upper epidermis) [139], and *CCHCR1* (associated variant WWCC; a widely expressed protein that is overexpressed in psoriasis) [140]. While exact identification has been technically challenging due to strong linkage disequilibrium, most studies agree that HLA-Cw6 is the susceptibility allele in PSORS1 [141]. Studies of HLA-Cw6 showed several important associations: overall increased risk of psoriasis, especially in white and Chinese populations [141,142]; early-onset and more severe disease, especially in those with positive family history [141,143]; guttate psoriasis and streptococcal pharyngitis, but not pustular psoriasis [143,144]. Other psoriasis susceptibility loci include PSORS2 on chromosome 17q24–q25, which spans the gene for caspase recruitment domain-containing protein 14 (CARD14), a scaffolding protein involved triggering NF-κB activation [145,146]; PSORS4 on chromosome 1q which spans the epidermal differentiation complex [147]; PSORS6 on chromosome 19p13, which spans the gene for tyrosine kinase 2 (TYK2) [148]; PSORS7 on chromosome 1p, which spans the gene for IL-23 receptor (IL-23R) [149].

Genome-wide association studies (GWAS) have largely confirmed the findings from previous linkage studies, including the importance of the PSORS1 and the MHC locus [150,151,152], while identifying additional risk loci through high resolution analyses of small nucleotide polymorphisms (SNPs) in large sample populations [153,154]. Many of these variants highlight the importance of the immune system in the pathogenesis of psoriasis [155,156]. Numerous GWAS studies identified variants in cytokines, receptors, and signaling pathways that are critical for Th17 cell function: IL-23R (the receptor that binds to IL-23 and promotes activation and expansion of Th17 cells); IL-12B (the shared p40 subunit of IL-23 and IL-12); STAT3 (a signal transducer in the JAK/STAT signaling pathway downstream of several cytokine receptors including IL-23R); and Runx1 (a transcription factor important in Th17 cell differentiation) [153,155,156,157,158]. Additional variants were identified in the NF-κB signaling pathway [159,160]. More recently, next-generation sequencing and expression profiling have been combined with linkage and association studies to identify genetic variants with high accuracy, as evidenced by the successful identification of loss-of-function mutation in IL-36 antagonist (IL-36N) as the genetic basis for generalized pustular psoriasis [161,162].

### 3.2. Treatments

#### 3.2.1. Topical Treatments

As with AD, topical therapies are the first-line treatment for psoriasis, both as monotherapy in mild-to-moderate disease and as combination therapy with oral systemic, biologic, and phototherapy in moderate-to-severe disease. Historically, treatment for psoriasis was limited to topical preparations of anthralin (also known as dithranol), which induces apoptosis of keratinocytes through production of reactive oxygen species, and coal tar, which inhibits production of IL-15 and nitric oxide [163,164]. Though still used today, these therapies have largely been replaced by TCS. Numerous RCTs demonstrated clinical efficacy for both continuous and intermittent treatment of psoriasis with varying potencies of TCS [165,166]. Topical vitamin D analogs (including calcitriol, calcipotriene, and tacalcitol) have broad effects of keratinocyte proliferation, apoptosis, and immunomodulation [167]. Their activity is comparable to mid-potency TCS (without the associated AEs), and is enhanced when combined or alternated with TCS [168,169]. Topical vitamin D analogs have few AEs, and systemic effects on vitamin D, calcium, and parathyroid hormone are extremely rare [170]. The TCIs tacrolimus and pimecrolimus are used off-label to treat psoriasis, particularly in the facial, genital, and intertrigenous areas [171]. While safe and efficacious, their use is limited by their overall potency, which is rated lower than mid-potency TCS and vitamin D analogs [172]. Topical retinoids (vitamin A derivatives) bind to the retinoic acid receptor (RAR) on both keratinocytes and non-immune cells, altering expression of genes important for differentiation, proliferation, and inflammation [173]. Tazarotene was the first retinoid developed for treatment of psoriasis, and, while efficacious, its continuous use is limited by local irritation. In combination with TCS, tazarotene shows increased potency and decreased local AEs [174,175].

#### 3.2.2. Phototherapy

Phototherapy is a therapeutic option for moderate-to-severe psoriasis involving a large BSA (>10%), both as monotherapy and combination therapy. Its utility, however, is practically limited by the requirement to travel to a phototherapy center multiple times weekly or own a home phototherapy unit. The mechanism of phototherapy is believed to be multifactorial, including induction of apoptosis in keratinocytes, APCs, and Th17 cells, and promotion of regulatory T-cell (Treg) activation [176]. Broadband UVB (290–320 nm) was used for nearly a century; however, it has largely been replaced by narrow band UVB (NB-UVB; 311 nm) and excimer laser (308 nm) [177]. NB-UVB has a good safety profile with a low risk for cutaneous malignancy. In contrast, an alternative to UVB-based therapy is psoralen combined with UVA (PUVA). Oral ingestion of 8-methoxsalen is followed by exposure to UVA light, resulting in selective apoptosis of proliferating cells. Once the gold standard for targeted phototherapy, PUVA has fallen out of favor due to its carcinogenic effects and association with squamous cell carcinoma and melanoma of the skin [178,179].

#### 3.2.3. Systemic Therapy

As was previously discussed for AD, systemic immunosuppressants, including methotrexate (the first systemic therapy approved for psoriasis) and cyclosporine A, have shown efficacy in treating moderate-to-severe psoriasis, but are limited by their relative non-specificity and broad AE profile. Acitretin is an oral second-generation retinoid that is approved for the treatment of moderate-to-severe psoriasis and is particularly efficacious for pustular psoriasis, erythrodermic psoriasis, and as an adjunct to phototherapy [180]. Similar to the mechanism of action of tazarotene, it binds to the RAR and regulates transcription of genes important for epidermal proliferation, keratinocyte differentiation, and immune cell production of cytokines. Overall, it appears to be less effective than traditional systemic agents for plaque psoriasis, and has several important AEs including dry skin and mucous membranes, dyslipidemia, elevated liver function tests, and potential teratogenicity [181].

#### 3.2.4. Phosphodiesterase-4 Inhibition

In 2014, the FDA approved apremilast for the treatment of moderate-to-severe psoriasis. As discussed earlier, apremilast is a small molecule inhibitor of the PDE-4 enzyme that reduces the level of pro-inflammatory Th17 cytokines and increases expression of IL-10 [182]. In multiple phase III double-blind, placebo controlled RCTs, patients with moderate-to-severe plaque or palmoplantar psoriasis treated with apremilast showed significant improvement in PASI-50 (Psoriasis Area and Severity Index, ≥50% reduction from baseline), PASI-75, and static investigators global assessment (sIGA) at week 16 [183,184,185,186]. AEs include nausea, diarrhea, and weight loss, which can be quite significant and chronic in some patients. Though apremilast does not carry the same immunosuppression risks as biologic therapy, it is overall less efficacious with only approximately 30% of patients achieving PASI-75 scores at week 16.

#### 3.2.5. TNF-α Inhibition

TNF-α inhibitors are the first generation of biologic medications for psoriasis, and they have been successfully used to treat moderate-to-severe psoriasis for over a decade. TNF-α is a pleiotropic, pro-inflammatory cytokine involved in both the innate and adaptive arms of the immune response in psoriasis. However, given its more general role in the immune response, TNF-α inhibitors have a broad AE profile, especially with regard to increased infectious risk (e.g., conversation of tuberculosis from latent to active, reactivation of hepatitis B virus). Etanercept (a recombinant human fusion protein consisting of the TNF-α receptor protein and Fc portion of immunoglobulin IgG1) was first-in-class and approved by the FDA in 2004, followed by infliximab (a human-murine chimeric monoclonal IgG1 antibody against TNF-α) in 2006, adalimumab (the first fully human monoclonal IgG1 antibody against TNF-α) in 2008, and certolizumab (a recombinant humanized IgG1 antibody Fab fragment against TNF-α conjugated to polyethylene glycol) in 2018 [187]. Phase III double-blind, placebo controlled RCTs for etanercept (PASI-75 at week 10–16: 47–49%) [188,189,190], infliximab (75–80%) [191,192], adalimumab (53–80%) [193,194,195,196], and certolizumab (67–83%) [197] demonstrated that biologic therapy with TNF-α inhibitors is efficacious and safe, thus paving the way for additional generations of immunotherapy.

#### 3.2.6. IL-12/IL-23 Inhibition

The FDA approved Ustekinumab for treatment of psoriasis in 2009. Representing the next generation of biologic therapy after TNF-α inhibitors, ustekinumab is a fully human IgG1 monoclonal antibody against the shared p40 subunit of IL-12 and IL-23, thus blocking two distinct pathways of T-cell activity: Th1 and Th17, respectively [187]. Th17 cells, and to a lesser extent Th1 cells, are crucial for orchestrating the chronic, maintenance phase of psoriasis, and, as such, the p40 subunit presents a more specific target for psoriasis compared to TNF-α. Phase III double-blind, placebo controlled RCTs showed a PASI-75 of 65–78% at week 12 with overall good efficacy, safety, and long-term drug survival [198,199,200]. While infectious complications were the most serious AEs associated with ustekinumab, the overall rate of infections was lower than that seen with TNF-α inhibitors [201].

#### 3.2.7. IL-17 Inhibition

Highlighting an evolving understanding of the immunological basis of psoriasis, the third generation of biologic medications approved by the FDA was the IL-17 inhibitors: secukinumab (a fully human monoclonal IgG1 antibody against IL-17A) in 2015; ixekizumab (a humanized monoclonal IgG4 antibody against IL-17A) in 2016; brodalumab (a human monoclonal IgG2 antibody against IL-17RA) in 2017 [187]. IL-17 is one of the signature cytokines produced by Th17 cells and responsible for amplifying the inflammatory cascade during the chronic phase of psoriasis. Phase III double-blind, placebo controlled RCTs showed high PASI-75 scores at week 12 (secukinumab: 75–87%; ixekizumab: 87–90%; brodalumab: 83–86%) and even significant, measurable PASI-90 and PASI-100 responses compared to placebo, TNF-α inhibitors, and/or ustekinumab [202,203,204,205,206,207]. Interestingly, even after discontinuation of treatment with secukinumab, approximately 20% maintained their response even after one year, suggesting the possibility of altering homeostatic set points with biologic therapy. Overall, while IL-17 inhibitors were well-tolerated and safe, AEs included upper respiratory infections, nasopharyngitis, and candidal infections, underscoring the important role of Th17 cells in extracellular defense at epithelial surfaces. However, the role of IL-17 inhibition in flaring inflammatory bowel disease (IBD) in those with a personal or family history of IBD is currently under investigation [208].

#### 3.2.8. IL-23 Inhibition

Based on clinical data from ustekinumab patients in conjunction with basic studies showing protective effects of IL-12 in psoriasis [209] and a more important role for IL-12 in Th1 mediated host defense, IL-23 was identified as a more specific target for psoriasis therapy [210]. The latest generation of biologic medications approved by the FDA for use in moderate-to-severe psoriasis are the IL-23 inhibitors: guselkumab in 2017; tildrakizumab in 2018; and risankizumab in 2019, all of which are human monoclonal IgG1 antibodies against the p19 subunit of IL-23 [187]. Primary or secondary endpoints for the phase III double-blind, placebo controlled RCTs of the IL-23 inhibitors were measured with PASI-90 (guselkumab: week 16, 70–85%; tildrakizumab: week 12, 35%; risankizumab: week 16, 72–75%) and PASI-100 scores, representing a remarkable improvement in treatment specificity [211,212,213,214,215,216]. IL-23 inhibitors showed higher efficacy compared head-to-head with TNF-α inhibitors and/or ustekinumab. As was seen with IL-17 inhibitors, about one-quarter of patients who discontinued therapy with risankizumab after 16 weeks maintained their response even after 48 weeks, again suggesting the possibility for long-term modification of cytokine signaling pathways with biologic therapy. IL-23 inhibitors were well-tolerated and safe, with most AEs related to infection (upper respiratory infection, gastroenteritis, herpes simplex virus, tinea).

#### 3.2.9. Janus Kinase Inhibition

Many of the Th17 and Th1 cytokines involved in the pathogenesis of psoriasis signal through JAK. Oral tofacitinib is currently an FDA-approved therapy for psoriatic arthritis [217,218], and has shown promising findings for the treatment of psoriasis in phase III double-blind, placebo controlled RCTs (PASI-75 at week 16: 40–59%) [219,220]. Topical tofacitinib has also demonstrated efficacy in phase II RCTs for psoriasis [221,222]. However, systemic tofacitinib use is associated with a number of infectious AEs (several of which were observed during trials), and, in 2015, the FDA declined to approve oral tofacitinib for use in moderate-to-severe without additional safety studies. Further investigation of tofacitinib, as well as other oral JAK inhibitors including baricitinib [223] and peficitinib [224], is ongoing.

## 4. Management in Children

For the treatment of mild-to-moderate AD or psoriasis in children, first-line therapies include TCS and TCI, with the addition of emollients for AD and topical vitamin D analogs for psoriasis. Data supporting their use in monotherapy and/or combination therapy is derived from years of anecdotal clinical experience and studies performed in adults, as RCTs are for these agents are rare in children, especially under the age of 12. TCS use in children is sometimes limited given concern for cutaneous (e.g., atrophy, fragility, striae, infection) and systemic (e.g., hypothalamic–pituitary axis suppression) AEs. While TCIs do not cause atrophy or hypothalamic–pituitary axis suppression, they are associated with application site reactions such as burning or stinging and a theoretical black-box warning by the United States Food and Drug Association for malignancy. Despite these potential AEs, both TCS and TCIs can be used safely and efficaciously in continuous and/or intermittent fashion. Adjunctive treatment with topical vitamin D analogs in children with psoriasis is overall very well tolerated with minimal cutaneous AEs and extremely rare systemic AEs (e.g., hypercalcemia, hypercalciuria) with overuse. Crisaborole was recently added as an additional first-line topical option in AD, and the combination of its relatively specific mode of action, non-steroidal formulation, and promising long-term safety data make it an important addition to the pediatric AD arsenal [225]. However, further studies are needed to better understand crisaborole’s efficacy in children, especially in comparison with TCS and TCIs. Common AEs of crisaborole in children include application site burning and/or stinging and, rarely, application site skin infection. For psoriasis, Tazarotene is rarely used in children given the associated skin irritation, while anthralin and coal tar are limited by practical features associated with application: staining of skin and clothing (both agents) and malodor (coal tar).

For the treatment of moderate-to-severe AD or psoriasis in children, treatment options include systemic therapy (most of which are used off-label), phototherapy, and biologic agents. Current guidelines recommend a stepwise approach to treatment in children with moderate-to-severe disease with biologic agents at the top of the therapeutic ladder. Methotrexate is the most frequently used systemic agent in children and has demonstrated reasonable efficacy in prospective and retrospective analyses, though the overall quality of data is low with high inter-study variability [226,227,228,229,230,231]. Disadvantages include the need for continuous lab monitoring and potential AEs including hepatotoxicity and liver fibrosis (especially with long-term use), hematologic abnormalities, and gastrointestinal symptoms. Cyclosporine A, which is usually reserved for flaring and/or refractory AD and psoriasis, has shown good efficacy in the short-term in several uncontrolled studies; however, there are sparse data on long-term effects in children [229,232]. Furthermore, chronic use of cyclosporine A is associated with a number of severe AEs including infection, malignancy, hypertension, nephrotoxicity, and hepatotoxicity. Acitretin can be effective for children with psoriasis (especially the pustular subtype) but can cause severe dryness of the skin and mucous membranes as well as dyslipidemia [229]. Other systemic therapies have limited data to support use, including mycophenolate mofetil and azathioprine. Both are associated with similar AEs that restrict their use in children, including diarrhea, nausea, vomiting, anemia, leukopenia, immune suppression, and increased risk of infection. Systemic corticosteroids are generally contraindicated for management of AD and psoriasis despite continued inappropriate use especially in childhood AD [233], and they should only be used as bridge therapy, for immediate relief, or when other options are not available. Common side effects of chronic systemic corticosteroid use in children are similar to those in adults and include immune suppression, hyperglycemia, weight gain, insomnia, gastroesophageal reflux, osteonecrosis, and adrenal insufficiency.

For both AD and psoriasis, phototherapy with either NB-UVB or excimer laser can be preferable to systemic immunosuppressive therapy in children given the limited AEs. Uncontrolled studies support its efficacy; however, multiple sessions per week in-office can be challenging for children and caregivers, and there continues to be concern about long-term exposure to UVB light starting in childhood [234,235,236].

Dupilumab is currently the only biologic agent available for the treatment of AD, though many new biologics are under investigation. The FDA recently expanded dupilumab’s indication in 2019 for use in children ≥12 years old with AD, and investigation for use in infants and children with AD is currently ongoing. Long-term safety and efficacy data for dupilumab use in children are eagerly anticipated, with adult open-label extension data supporting its continuous, long-term use [237]. The most common treatment-related AEs observed in both adolescents and adults included injection site reactions, conjunctivitis, and nasopharyngitis. For children with psoriasis, etanercept [238], adalimumab [239], and ustekinumab [240] have been studied in phase III double-blind, placebo controlled RCTs. All three agents showed comparable efficacy and safety to the previous adult trials. Adalimumab showed higher efficacy in head-to-head comparison with methotrexate. Currently, the only FDA-approved biologics in children are etanercept (≥4 years old) and ustekinumab (≥12 years old), with ongoing studies for many of the newer treatment classes. In general, biologic treatment-associated AEs in children with psoriasis are similar to those observed in adults and more favorable compared to classic systemic therapies. Common AEs associated with TNF-α and IL-12/IL-23 inhibitors use in children include injection site reactions, nasopharyngitis, mild upper respiratory infection (similar to those normally seen in children), and headache.

## 5. Conclusions

Improved understanding of the pathophysiology of AD and psoriasis has led to a revolution in targeted therapy. Targeted treatments have filled many unmet needs for both of these conditions. As additional novel agents are approved, the next goal in management of AD and psoriasis is to optimize specific treatments for individual patients. The next decade of research will be dominated by extensive genetic, molecular, and clinical phenotyping of patients in order to understand which pathologic mechanisms are most relevant. The era for personalized medicine in AD and psoriasis is upon us!

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
