# Peer review of "Pathophysiology of Atopic Dermatitis and Psoriasis: Implications for Management in Children"

_children, 2019, doi:10.3390/children6100108_

Round 1

Reviewer 1 Report

The manuscript concisely but sufficiently covers pathophysiology and relevant treatments for AD and psoriasis. Considering the common age of onset is quite different for both diseases, brief explanation about the epidemiological features of both diseases, specifically in children, is recommended. 

Author Response

In response to Reviewer #1, we have added additional details about epidemiologic factors of atopic dermatitis and psoriasis as highlighted in sections 2 and 3. We have overall tried to narrow the scope of this review to pathophysiology and treatment and limit our discussion of epidemiology out of respect for the other articles in this special issue that will further expand upon this topic.

Reviewer 2 Report

Authors need to consider the evidence for and against use of these agents in children

Also potential risks of using these agents in children.

This is not explored in the manuscript.

This reads more as a text-book like review rather than an evaluation and critical appraisal of the literature, which would be of much more interest to readers

Author Response

In response to Reviewer #2, we have added additional details about evidence for use and risks of children-specific agents in section 4. Many of these agents have risk-benefit profiles that are also explained in detail in earlier sections of the review. Additionally, other agents do not have sufficient data that support their routine use in children, which we have also indicated in our text. We have written this review to provide an overview of pathophysiology and current treatment options that have been influenced by the current understanding of pathologic mechanisms in atopic dermatitis and psoriasis. Rather than focusing on our personal opinion of any specific agent or treatment protocol, we have tried to present a balanced view of the current therapeutic landscape while highlighting commonly used treatments in children, both approved and off-label. We have thus tried to focus the scope of our review in this manner, as there are additional articles in this special issue that will discuss treatment in great detail.

Round 2

Reviewer 2 Report

manuscript now reads well